# Ontogeny of the Respiratory Area in Relation to Body Mass with Reference to Resting Metabolism in the Japanese Flounder, *Paralichthys olivaceus* (Temminck & Schlegel, 1846)

**Dong In Kim**

Aquaculture Research Institute, Kindai University, Shirahama 3153, Nishimuro, Wakayama 649-2211, Japan; donginkim0508@gmail.com or dongin_kim@kindai.ac.jp

**Abstract:** Metabolism is the fundamental process dictating material and energy fluxes through organisms. Several studies have suggested that resting metabolic scaling in various aquatic invertebrates is positively correlated with changes in body shape and the scaling of body surface area, which agrees with the surface area theory, but contradicts the negative correlations predicted by the resource–transport network theory. However, the relationship between resting metabolic scaling and respiration area, particularly in asymmetric fish that have undergone dramatically rapid metamorphosis, remains unclear. In this morphometric study in an asymmetric fish species (*Paralichthys olivaceus*), I compared my results with previous reports on resting metabolic scaling. I measured the respiratory area of *P. olivaceus* specimens aged 11–94 days (body weight, 0.00095–1.30000 g, respectively) to determine whether and how the resting metabolic scaling is associated with changes in body shape and respiratory area. Resting metabolic scaling might be more closely related to body surface area, because their slopes exactly corresponded with each other, than to respiratory area. Furthermore, confirming the surface area theory, it was linked to changes in body shape, but not from the resource–transport network theory. These findings provide new insights into the scaling mechanisms of area in relation to metabolism in asymmetric fish.

**Keywords:** fish morphology; teleost; asymmetric fish; body shape; respiratory organ; size-scaling metabolism; gill area; cutaneous area; oxygen consumption

## 1. Introduction

In many organisms, oxygen consumption is considered a reflection of metabolism, which is the fundamental process by which food is changed into energy. Although metabolism may affect optimal resource allocation to life processes such as growth and bodily maintenance, the reverse may also occur [1–7].

Metabolism, also denoted as $VO_2$, can be expressed as a power function of the body mass (M) of an organism according to the allometric equation $VO_2 = aM^b$, where a is the scaling coefficient (the antilog of the intercept in a log–log plot) and b is the scaling exponent (the slope in a log–log plot) [8,9]. The $VO_2$ of an organism is generally accepted to show negative allometry, scaling to body mass, because the scaling exponent is less than unity. Thus, the mass-specific rate of metabolism ($VO_2/M$) should decrease as an organism's body mass increases [9–18]. For decades, however, predictions of scaling exponent values have been continually challenged, and this remains a highly debated topic in physiology and ecology [6,19–30].

The scaling coefficient is species-specific, but the scaling exponent for resting metabolic rate (i.e., the standard metabolic rate of ectotherms or the basal metabolic rate of endotherms) has, over many years, been commonly given as approximately three-quarters (3/4), a generalization known as "Kleiber's law" or "the 3/4-power law" [10,13,14,31–36]. Nevertheless, contrasting theories have recently arisen because of several reported exceptions in which the resting metabolic rate of the scaling exponent was significantly different

from the universal 3/4-power law [20–23,26,37–45]. Deviations from the 3/4-power law are known to occur interspecifically (ranging from approximately 0 to >1, but mostly found between 2/3 [two-thirds] and 1) [20,21,23,26,46–49], as well as intraspecifically [15,23,26].

Several theoretical approaches have been taken to explain the extensive variation in the metabolic scaling exponent, and the subject remains under debate [12,13,44,50]. Two competing theories have become prominent in the study of metabolic rates based on geometrical relationships between body mass and resource supply [45,51]. These theories respectively suggest that metabolic scaling is strongly dependent on the transport of materials through external exchange surfaces such as those of respiratory organs (the "Surface Area" or SA theory) [23,41,45,51–53] or the geometrical properties of internal networks (the "Resource Transport Network (RTN) theory) [37,39,54,55]. According to both theories, changes in metabolic rate during ontogeny are affected by scaling with body mass; however, in each theory, the changes in body shape have opposite effects on the metabolic scaling exponent [45,51], i.e., if an organism displays elongated one-dimensional or flattened two-dimensional growth during ontogeny, the SA theory predicts that the metabolic scaling exponent will increase (isometrically with body mass, i.e., "b" is equal or close to 1) [45,53], whereas the RTN theory predicts that the scaling exponent will decrease (showing negative allometry with body mass, i.e., "b" is < 2/3) [37,39,40,45]. Conversely, if an organism shows isomorphic three-dimensional growth during ontogeny, the SA theory predicts that the metabolic scaling exponent will decrease (showing negative allometry with body mass, i.e., "b" is close to 2/3) [51], whereas the RTN theory predicts that the scaling exponent will increase (showing negative allometry with body mass, i.e., "b" is > 2/3) [37,39,40,54] depending on network properties [39,40,42,55,56].

In several previous studies, the scaling of surface area with body mass has been geometrically estimated in various aquatic invertebrates, and these estimates significantly positively correlated with elongated or flattened body shape during ontogeny, as predicted by the SA theory, whereas the estimates predicted using the RTN theory included incorrect negative correlations [45,51,57]. These theoretical approaches have been applied to study multiple levels of various biological processes; specifically, however, the scaling exponent of surface area in relation to body mass could influence the body surface-related material exchange capacity that is required for metabolism [45,51,57]. Previous studies have demonstrated that the ontogenetic phase shifts in the metabolism of animals, including fishes, occur as body mass is increased during their early life history [4,23,58,59]. Interestingly, the transitional phases of metabolism are accompanied by not only morphological changes but also behavioral and/or environment-related changes during the development stages of animals [4,59]. Thus, these changes can serve as accurate indicators of aquatic ecological implications as well as the feeding, growth, and environmental requirements of aquatic organisms [1–5,7,44,51,57].

Several studies involving fish have reported that the scaling of the surface area–body mass relationship is related to the scaling of metabolic rate; however, this phenomenon is yet to be fully explained, because most of these studies focused on limited stages in the life history of specific organisms [60–70]. Although several studies have examined how the area of respiratory surfaces correlates with metabolic rate and scaling according to body size and age in various aquatic animals [45,51,57,71–79], additional research is required to determine how changes in body shape during rapid metamorphosis in early life stages affect the scaling of surface area and metabolic rate.

Japanese flounder, *Paralichthys olivaceus*, is an optimal species for conducting such studies. Yagi et al. [59] reported the ontogenetic phase shifts in metabolism of this species during its early life history, i.e., the allometric relationships between $VO_2$ and M ranging from 0-day-old larvae weighing 0.00026 g to 80-day-old juveniles weighing 0.90000 g. They found three interposing transitional phases of $VO_2$ at approximately 0.002, 0.010, and 0.200 g, with the scaling exponent kept constant in each phase (b = 0.831). Interestingly, when *P. olivaceus* undergo explosive metamorphosis during the larval to juvenile stages, their body shape is completely changed from a symmetrical to an asymmetrical form, and

their swimming and/or environment-related behavior changes from pelagic (horizontal positioning) to benthic (bottom-dwelling) habits [59,80,81].

Taking these findings together, I hypothesized that the metabolic scaling of asymmetric fish, in this case the Japanese flounder, would be strongly influenced by changes in body shape with respiratory surface area during rapid metamorphosis. Therefore, morphometric studies on fishes from their larval to juvenile stages (i.e., those undergoing metamorphosis) must be conducted. It is important to determine whether and how the scaling of metabolic rate is associated with changes in the body shape and respiratory surface area. In the present study, I conducted such morphometric analyses of the allometric relationships of the respiratory surface area of Japanese flounder in relation to body mass; my tests included a range of developmental stages from 11-day-old larvae weighing 0.00095 g to 94-day-old early juveniles weighing 1.30000 g. Furthermore, I also discuss the ontogenetic phase shifts in metabolism in relation to the development of the respiratory organs.

## 2. Materials and Methods

### 2.1. Fish Specimens

The metabolic rate of fishes varies depending on environmental factors. In particular, water temperature has a direct effect on metabolism [48]. Therefore, for comparisons to be valid, experiments should be conducted under similar water temperatures. In a previous study, resting metabolic rate was measured under a water temperature of approximately 18 °C, which is an optimal temperature during the early development stages of *P. olivaceus* [59]. In the present study, *P. olivaceus* specimens were obtained using in vitro fertilization and sampled as 11-day-old larvae weighing 0.00095 g up to 94-day-old early juveniles weighing 1.30000 g. Originally, the fish were provided by the local fishermen's association at the northern part of Kyushu, Japan, and transported to the Fisheries Research Laboratory, Kyushu University, Fukuoka, Japan. All larvae and juveniles were maintained in 500-L polycarbonate cylinder tanks at approximately 18 °C with a constant flow of seawater. They were fed on the S-type rotifer *Brachionus rotundiformis* (cultured by supplying the condensed freshwater Chlorella at 28 °C) twice daily from 2 to 20 days after hatching; with brine shrimp, *Artemia* sp., larvae (hatched during 24 h incubation in sea water at 28 °C) twice daily from 18 to 40 days after hatching; and with an artificial diet accompanied by live brine shrimp and rotifers thrice daily thereafter. Diets containing live organisms were enriched with essential fatty acids, EPA, and DHA using Super Capsule Powder (Chlorella Industry, Tokyo, Japan) before feeding.

### 2.2. Measurement of Various Body Areas

Gill development was observed every day until 10 days after hatching to track the development of the early gills; however, these data were not used as direct surface area data. Gill, body surface, and fin areas were measured for 22 individuals weighing 0.00095–1.30000 g (from 11-day-old larvae to 94-day-old early juveniles). Before body-mass determination and/or morphological observation, fish were carefully anesthetized using seawater at 2–3 °C. The body mass of the small-sized fish, especially of those at early larval stages, was measured according to the following process. First, the weight of the cover glass to be used in histologic sections was measured using a high-precision electronic balance. Second, larvae were collected from the beaker in a few milligrams of seawater using a pipette and then carefully placed onto the cover glass. Third, excess seawater was carefully removed using a small piece of filter paper to enable the accurate measurement of larval body mass. Finally, the larval body mass was determined by subtracting the weight of the cover glass from that of the larvae and cover glass.

After larval body mass was determined, the larvae were either fixed using 2.5% glutaraldehyde fixative for 12–24 h and preserved in 70% ethanol or fixed whole in formalin Cortland's saline comprising one part concentrated formalin and nine parts Cortland's saline [82]. Outlines of the fish specimens were drawn under binocular microscope observation using an attached camera lucida, after which the gills of the specimens were

extracted. The gill area (GA) of the secondary lamella were determined in mm$^2$ according to the method of Hughes [83,84] as follows:

$$GA = (2L/d') \times bl,$$

where L is the total length of all the filaments (mm), $1/d'$ is the average spacing of the secondary lamellae on the side of the filaments (mm$^{-1}$), and bl is the average bilateral lamellar area (mm$^2$) exposed to the external medium.

### 2.3. Filament Length

The total length of all the filaments (L) was calculated by doubling the values of every filament length obtained from all gill arches on the left side of the body in all fish.

### 2.4. Spacing

The spacing of the secondary lamellae $(1/d')$ on one side of the filaments was usually determined using estimated values from all the lamellae on the filaments of average length in the dorsal, middle, and ventral parts of the second gill arches on the left side of the body. In small fish, i.e., those weighing <0.002 g, the spacing was determined from measurements of all the lamellae on the filaments of all the gill arches on the left side of the body, because their secondary lamellae on the second gill arches were not yet developed. The average spacing was calculated using the weighted mean method, which takes into account the length differences among different filaments and/or regions on a filament [74,75].

### 2.5. Lamellar Area

The average bilateral area of the secondary lamellae (bl) was determined using the triangle method previously used by Price [60]. This method can be applied to calculate the average bilateral area of the secondary lamellae by multiplying the average maximum height (b) and the average base length (l) of the lamellae while regarding them as a triangle structure. The detailed measurement methods are described below. To determine the average maximum height of the lamellae, the lamellae on the filaments of average length were estimated using the average for the dorsal, middle, and ventral parts of the second gill arches on the left side of the body. The number of lamellae determined was 4–51 in fish weighing 0.00095–1.30000 g. The average base length of the lamellae was estimated by measuring the lamellae on the filaments of average length in the dorsal, middle, and ventral parts of the second gill arches on the left side of the body. The number of secondary lamellae measured was 2–348 in fish weighing 0.002–1.300 g.

### 2.6. Measurement of Cutaneous Area

Body surface area was estimated by multiplying body length by the average circumference obtained from the body heights and body widths of ten evenly spaced parts according to the parabolic equation of Oikawa and Itazawa [74,75], which assumes that the circumference is composed of two parabolas:

$$C = \sqrt{4H^2 + B^2} + \frac{B^2}{2H} \times \ln\frac{2H + \sqrt{4H^2 + B^2}}{B},$$

where C is circumference length, H is body height, and B is body width. The real estimated values were obtained from corrected values calculated by comparing the estimated values between unfixed-fresh samples and glutaraldehyde- or formalin-fixed samples. Fin area was calculated by tracing the outline of the magnified specimen under a binocular microscope with an attached camera lucida. After using these approaches, the estimated values were multiplied by 1.021 (for glutaraldehyde) or 1.027 (for formalin) for conversion into real estimated values.

In the larval stage were specimens weighed <0.001 g (about 10 days after hatching), the boundaries between body surface area and fin area were unclear, except for a few

sections, such as the pectoral fins, because the subdermal space made it difficult to separate the fin folds from the body surface. Thus, body surface area and fin area were regarded as the same area during this developmental stage. Body surface area including fin folds was measured using the parabolic equation given above; in this case, H was body width and B was body height including the fin folds.

### 2.7. Data Analysis

Surface area–body mass analyses were performed using the statistical model described by Oikawa et al. [75]. Briefly, the relationships of surface area (SA) to body mass (M), i.e., $SA = \alpha M^{\beta}$, were classified as negative ($\beta < 1$) or positive ($\beta > 1$) allometry, where $\beta$ differed significantly ($p < 0.05$) from unity. Allometry with $\beta$-values that did not differ significantly from unity was classified as isometry ($\beta = 1$). Diphasic and/or triphasic allometry was distinguished via analysis of covariance (ANCOVA) using Microsoft Excel (significance level: $p < 0.05$).

## 3. Results
### 3.1. Gill Development

The morphological changes observed on the gills are described in Figure 1. Briefly, the fourth gill arches were first observed in 11-day-old larvae (0.00095 g in body mass; 4.75 mm in body length). In this developmental stage, gill filaments also appeared on the anterior and/or posterior hemibranches on the first to third gill arches, although secondary lamellae had not yet appeared. Secondary lamellae first appeared in 17-day-old larvae (0.002 g in body mass; 6.09 mm in body length). Gill filaments on the fourth gill arches largely developed in 30-day-old larvae (0.01125 g in body mass; 10.48 mm in body length). Secondary lamellae on the fourth gill arches had appeared in their entirety in 35-day-old larvae (0.02483 g in body mass; 12.92 mm in body length).

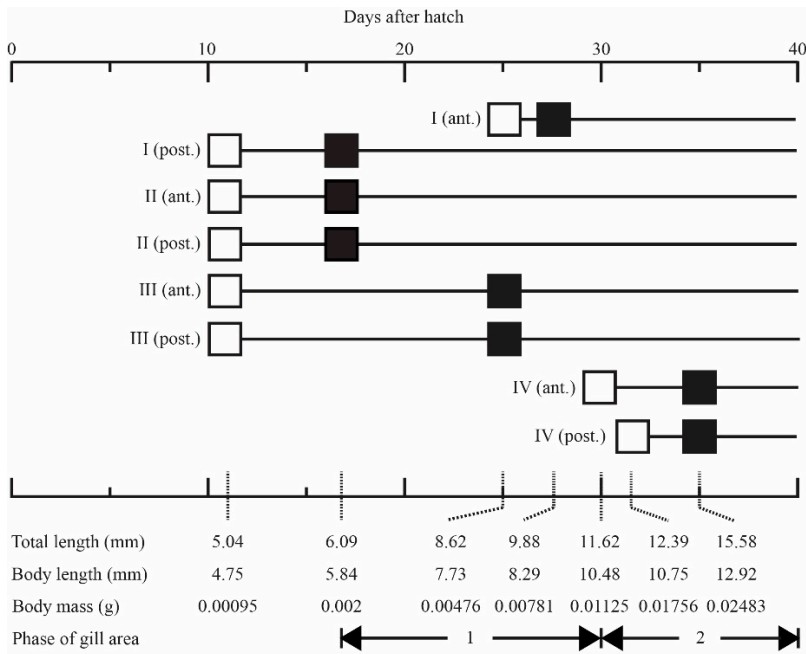

**Figure 1.** Schematic diagram of gill development in Japanese flounder. Square symbols indicate the first appearance of gill filaments (open black squares) and secondary lamellae (closed black squares) on the anterior (ant.) and posterior (post.) hemibranches of the first (I) to fourth (IV) gill arches.

### 3.2. Regression Analyses of Gill Area Measurements

Regression analyses of gill measurements are presented in Table 1. Fish in the larval stage (<11-day-old larvae at 0.00095 g) did not have a branchial area, because their secondary lamellae were not developed (Figure 1). Therefore, measurements of GA were

performed in specimens weighing 0.002–1.300 g (from 17-day-old larvae to 94-day-old early juveniles).

**Table 1.** Regression analyses of the allometric relationship ($Y = \alpha M^{\beta}$) between gill dimensions (Y) and body mass (M).

| Y | Range of Body Mass (g) | $n$ | $\alpha$ | $\beta$ ($\bar{x} \pm$ SE) | $r^2$ |
|---|---|---|---|---|---|
| Gill area (mm$^2$) | 0.00200–0.01125 | 7 | $8.6 \times 10^4$ | 2.197 ± 0.287 ** | 0.921 |
| | 0.01125–0.11665 | 7 | 1340.21 | 1.354 ± 0.079 ** | 0.983 |
| | 0.11665–1.30738 | 9 | 833.61 | 1.114 ± 0.074 | 0.970 |
| Total filament number | 0.00095–0.00555 | 6 | 1031.72 | 0.406 ± 0.136 * | 0.689 |
| | 0.00555–0.08122 | 7 | 2994.11 | 0.578 ± 0.061 *** | 0.947 |
| | 0.08122–1.30738 | 11 | 1189.48 | 0.265 ± 0.030 *** | 0.896 |
| Total filament length (L, mm) | 0.00095–0.00555 | 6 | 7100.39 | 1.165 ± 0.137 | 0.948 |
| | 0.00555–0.15100 | 11 | 1339.96 | 0.834 ± 0.035 ** | 0.984 |
| | 0.15100–1.30738 | 7 | 987.77 | 0.632 ± 0.064 ** | 0.951 |
| Spacing of the secondary lamellae (1/d', mm$^{-1}$) | 0.00200–0.00781 | 6 | 152.88 | 0.348 ± 0.137 ** | 0.618 |
| | 0.00781–0.11035 | 7 | 20.27 | −0.058 ± 0.039 *** | 0.308 |
| | 0.11035–1.30738 | 10 | 24.92 | −0.033 ± 0.068 *** | 0.045 |
| Average bilateral area of the secondary lamellae (bl, mm$^2$) | 0.00200–0.00555 | 5 | 1000.76 | 2.107 ± 0.229 * | 0.966 |
| | 0.00555–0.11665 | 9 | 1.35 | 0.873 ± 0.067 | 0.960 |
| | 0.11665–1.30738 | 9 | 1.41 | 0.864 ± 0.079 | 0.945 |

The difference of β-value from unity was examined by *t*-test. * $0.01 < p < 0.05$, ** $0.001 < p < 0.01$, *** $p < 0.001$. $r^2$ is squared correlation coefficient between logYand logM.

There was a triphasic allometric relationship between GA and body mass, with statistically significant differences among these relationships according to ANCOVA ($p < 0.05$): allometry was positive in the first to second phases, but it was isometric in the third phase (Figure 2 and Table 1).

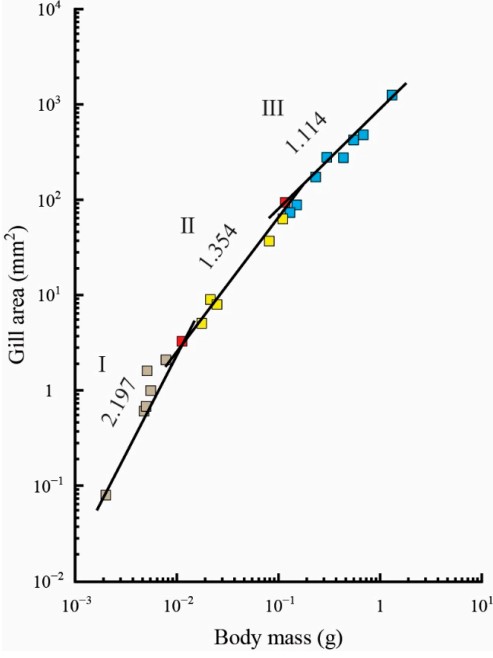

**Figure 2.** Allometric relationship between the gill area (colored squares) and body mass. The values above the regression lines indicate their slopes. The gill area was measured using the triangle method. The ontogenetic phase shifts in the gill area are presented as squares in three colors (gray, phase I; yellow, phase II; blue, phase III). Red-colored squares indicate the gill area measurements that overlap between phases I and II and between phases II and III.

The first phase (about 11–30 days after hatching; 0.002–0.010 g in body mass) corresponded to the morphological and behavioral changes in the early larval stage because the appearance of secondary lamellae and critical incremental swimming velocity were observed. The second and third phases were closely correlated with the transformation from the larval stage to the first half (about 31–50 days after hatching; 0.01–0.10 g in body mass) and second half (about 51–90 days after hatching; 0.1–1.3 g in body mass) of the juvenile stage, respectively (Figure 2 and Table 1).

These transitional patterns in allometric relationships generally corresponded to the changes, following an increase in body mass, in the total filament number, length, average spacing of the secondary lamellae on one side of the filaments, and average area of the secondary lamellae (Figure 3A–D and Table 1).

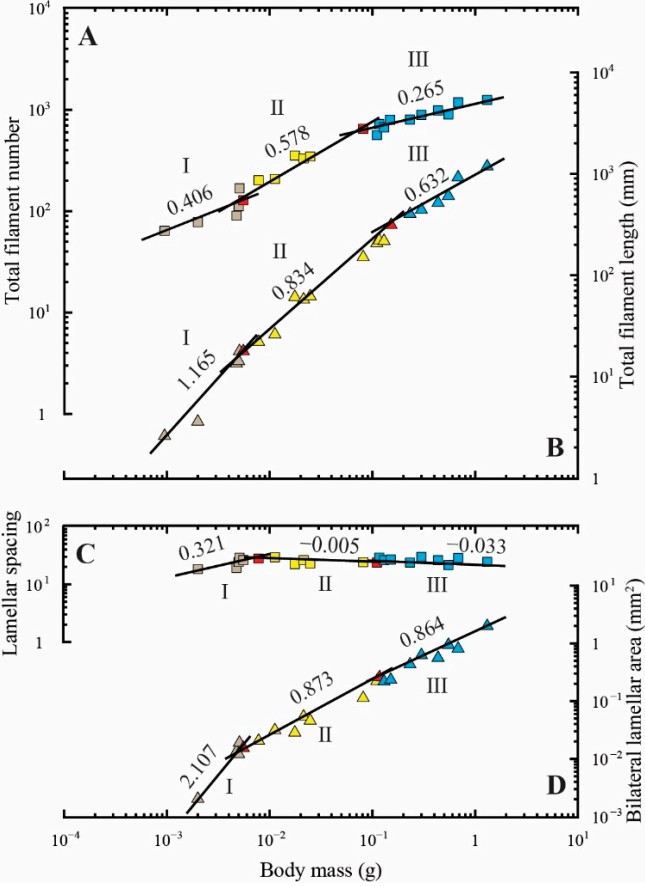

**Figure 3.** Allometric relationships between body mass and total filament number (colored squares) (**A**); total filament length (colored triangles) (**B**); average spacing of the gill secondary lamellae on one side of the filament (colored squares) (**C**); average bilateral lamellar area (colored triangles) (**D**). The values above the regression lines indicate their slopes. The ontogenetic phase shifts are presented in three colors (gray, phase I; yellow, phase II; blue, phase III). The red-colored spots indicate the measurements that overlap between phases I and II and between phases II and III.

### 3.3. Regression Analyses of Cutaneous Area Measurements

Regression analyses of the cutaneous measurements are provided in Table 2. Triphasic allometry existed between total cutaneous area and body mass ($p < 0.01$; Figure 4A). The allometry was negative in the first to third phases (Figure 4A and Table 2). The slopes of the total cutaneous area in the first to second phases (0.815 and 0.625, respectively) were nearly 2/3, whereas the slope in the third phase (0.873) was significantly larger than 2/3 ($p < 0.01$; Figure 4A and Table 2).

**Table 2.** Regression analyses of the allometric relationships ($Y = \alpha M^\beta$) between body mass (M) and body surface area, fin area, total cutaneous area, and total length (Y).

| Y | Range of Body Mass (g) | $n$ | $\alpha$ | $\beta$ ($\bar{x} \pm$ SE) | $r^2$ |
|---|---|---|---|---|---|
| Total cutaneous area (mm²) | 0.00095–0.00555 | 6 | 3784.48 | 0.815 $\pm$ 0.064 * | 0.976 |
| | 0.00555–0.11035 | 8 | 1383.61 | 0.625 $\pm$ 0.041 *** | 0.974 |
| | 0.11035–1.30738 | 10 | 2598.44 | 0.873 $\pm$ 0.059 | 0.964 |
| Body surface area (mm²) | 0.00095–0.00555 | 6 | 1135.94 | 0.676 $\pm$ 0.047 ** | 0.981 |
| | 0.00555–0.11665 | 9 | 850.97 | 0.623 $\pm$ 0.041 *** | 0.971 |
| | 0.11665–1.30738 | 9 | 1743.07 | 0.893 $\pm$ 0.089 | 0.936 |
| Bilateral fin area (mm²) | 0.00095–0.00555 | 6 | 395.15 | 0.574 $\pm$ 0.139 * | 0.810 |
| | 0.00555–0.11035 | 8 | 557.18 | 0.643 $\pm$ 0.064 ** | 0.944 |
| | 0.11035–1.30738 | 10 | 826.98 | 0.790 $\pm$ 0.062 ** | 0.953 |
| Total length (mm) | 0.00095–0.00555 | 6 | 65.32 | 0.373 $\pm$ 0.029 *** | 0.976 |
| | 0.00555–0.08122 | 7 | 39.33 | 0.276 $\pm$ 0.025 *** | 0.961 |
| | 0.08122–1.30738 | 11 | 69.08 | 0.488 $\pm$ 0.033 *** | 0.961 |

The difference of β-value from unity was examined by *t*-test. * $0.01 < p < 0.05$, ** $0.001 < p < 0.01$, *** $p < 0.001$. $r^2$ is squared correlation coefficient between logY and logM.

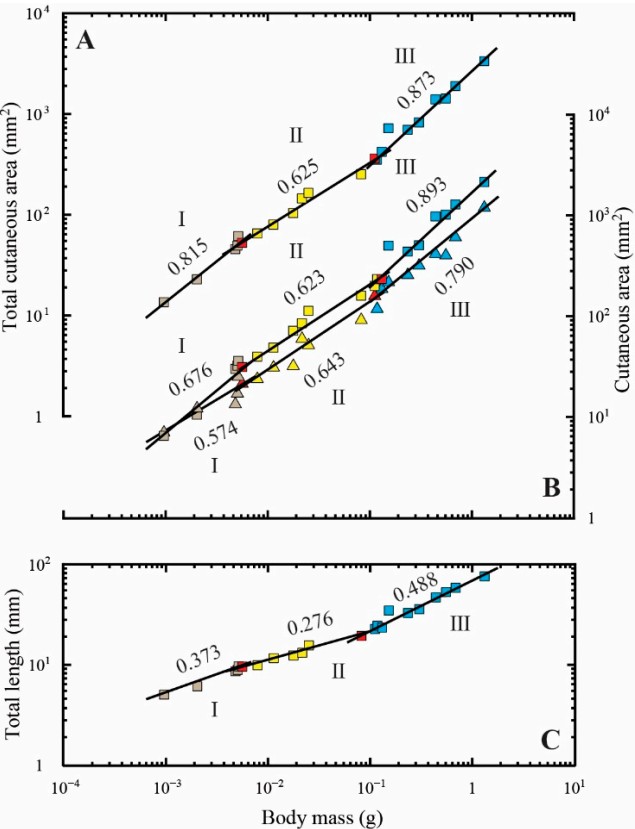

**Figure 4.** Allometric relationships between the body mass and total cutaneous area (colored squares) (**A**); body surface area (colored squares) (**B**); bilateral fin area (colored triangles) (**B**); total length (colored squares) (**C**). The values above the regression lines indicate their slopes. The ontogenetic phase shifts are presented in three colors (gray, phase I; yellow, phase II; blue, phase III). The red-colored spots indicate the measurements that overlap between phases I and II and between phases II and III.

In addition, triphasic allometry existed between body surface area and body mass ($p < 0.05$; Figure 4B and Table 2). The slopes of the body surface area in the first to second

phases (0.657 and 0.630, respectively) were nearly 2/3, whereas the slope in the third phase (0.893) was significantly larger than 2/3 ($p < 0.001$; Figure 4B and Table 2).

Triphasic allometry was also observed between fin area and body mass ($p < 0.05$) (Figure 4B and Table 2). Fin area increased with negative allometry and the slopes (0.574, 0.643, and 0.790, respectively) were nearly 2/3 (Figure 4B and Table 2).

The patterns for transitions in the allometric relationships of total length to body mass were closely correlated with changes following the relationship for total cutaneous area and body surface area (Figure 4A–C and Table 2). The slopes for total cutaneous area, body surface area, and total length in the first to second phases decreased with increasing body mass, whereas the slopes for fin area in the first to third phases gradually increased with increasing body mass (Figure 4A–C and Table 2).

### 3.4. Regression Analyses of Body Form Measurements

Regression analyses of body form are shown in Table 3. The relationships of body length to body mass, mean body height, and mean body width showed diphasic allometry ($p < 0.001$, $p < 0.05$, and $p < 0.01$, respectively; Figure 5 and Table 3).

**Table 3.** Regression analyses of the allometric relationships ($Y = \alpha L^{\beta}$) between body length (L) and body mass, mean body height, and mean body width (Y).

| Y | Range of Body Length (mm) | n | α | β ($\bar{x} \pm$ SE) | r² |
|---|---|---|---|---|---|
| Body mass (g) | 4.75–16.17 | 12 | $4.03 \times 10^{-6}$ | $3.500 \pm 0.139$ *** | 0.984 |
| | 16.17–66.30 | 11 | $3.67 \times 10^{-4}$ | $1.924 \pm 0.113$ *** | 0.970 |
| Mean body height (mm) | 4.75–10.75 | 9 | $1.57 \times 10^{-1}$ | $1.218 \pm 0.210$ ** | 0.827 |
| | 10.75–66.30 | 14 | $4.65 \times 10^{-1}$ | $0.831 \pm 0.025$ *** | 0.989 |
| Mean body width (mm) | 4.75–20.02 | 14 | $4.90 \times 10^{-2}$ | $1.139 \pm 0.057$ * | 0.971 |
| | 20.02–66.30 | 9 | $1.22 \times 10^{-1}$ | $0.847 \pm 0.041$ ** | 0.984 |

The difference of β-value from unity was examined by *t*-test. * $0.01 < p < 0.05$, ** $0.001 < p < 0.01$, *** $p < 0.001$. r² is squared correlation coefficient between logYand logM.

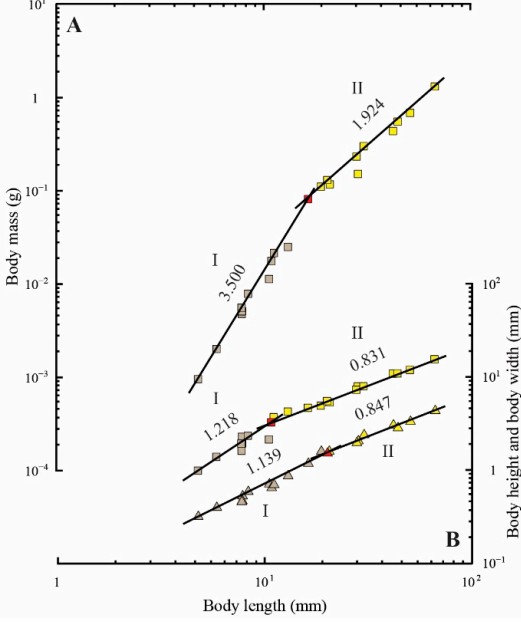

**Figure 5.** Allometric relationships between the body length and body mass (colored squares) (**A**); mean body height (colored squares) (**B**); mean body width (colored triangles) (**B**). The values above the regression lines indicate their slopes. The ontogenetic phase shifts are presented in two colors (gray, phase I; yellow, phase II). The red-colored spots indicate the measurements that overlap between phases I and II and between phases II and III.

The slopes of body mass and mean body width showed positive allometry in the first phases (3.500 and 1.139, respectively), whereas mean body height showed isometric allometry (1.218) (Figure 5 and Table 3). The slopes of body mass in the second phase (1.924) showed positive allometry, whereas mean body height and body width (0.831 and 0.847, respectively) indicated negative allometry (Figure 5 and Table 3). A notable inflective point of changing body form was found at approximately 10.75 mm (0.01756 g in body mass) to 20.02 mm (0.12968 g in body mass) in length, although the transitional patterns of body mass, mean body height, and mean body width were slightly different (Figure 5 and Table 3).

## 4. Discussion

### 4.1. Gill Morphometry

The results from morphometry analysis of the gills indicated that the GA of fish in the early larval stage, i.e., 0.002–0.010 g in body mass, rapidly increased with increasing body mass when compared with changes in later life stages (Figure 2 and Table 1). The rapid increase in GA was likely associated with increased oxygen demand, because the secondary lamellae do not develop in the early larval stage (<0.002 g in body mass; Figure 1). Such observations are a recurring feature in many bilaterally symmetric fish species. For example, early development of the GA has been characterized by rapid increases, with slopes of 3.36 in herring (*Clupea harengus*), 7.07 in common carp (*Cyprinus carpio*), 3.44 in rainbow trout (*Oncorhynchus mykiss*), and 2.22 in red porgy (*Pagrus major*) having been reported [68,74,75,85,86]. Similar to the abovementioned species, the slope for the relationship of GA was also steeper before metamorphosis than that after metamorphosis in the Japanese flounder studied here (Figure 2 and Table 1). Similar patterns were observed in the relationships between body mass and the total filament length, spacing of the secondary lamellar, and average bilateral area of the secondary lamellar (Figure 2 and Table 1). Nevertheless, Japanese flounder showed dramatic changes in asymmetrical form after metamorphosis that resulted in slightly different ontogenetic processes in the gill morphology compared with those in bilaterally symmetric fish. In particular, previous studies on red porgy have revealed that the total filament number shows diphasic negative allometry, with slopes of 0.620 and 0.197 calculated during the early larval (0.00034–0.00980 g in body mass) and later life stages (0.0098–1080.0000 g), respectively [75]. The rate of increase in the total filament number seems to constantly diminish during the early developmental stages. In contrast, our results indicate that the slopes were temporally increased from the first to second phase during the larval stage (at 0.00095–0.08122 g), after which they rapidly decreased during phasic transition from the second to third phase during the larval to juvenile stage (0.08122–1.30738 g; Figure 3A and Table 1).

The differences in the ontogenetic processes in gill morphology that exist between bilaterally symmetric and asymmetric fish are likely constrained by changes in their skeletal structures during metamorphic events. Even in bilaterally symmetric fish, gill morphological changes have been reported when skeletal deformity has occurred. Atlantic salmon (*Salmo salar* L.) can be used as an example of how skeletal deformity affects gill morphology [87]. Specifically, the rate of skeletal deformity is significantly higher in triploid salmon than that in diploid salmon. Their short opercula, gill filaments, and some jaw deformities show asymmetrical manifestations. Thus, skeletal deformity in bilaterally symmetric fish may cause the development of asymmetric skeletal structures while also affecting changes to gill morphology.

Similarly, Okada et al. [81] observed the development of cartilage and bone during the eye migration of metamorphosing Japanese flounder. At the onset of metamorphosis, the right eye begins to move into the dorsal margin when the fish is 7.40 mm in body length and weighs about 0.005 g (Figure 5A and Table 3), and the skeletal structures, such as the parasphenoid, trabecular cartilage, supraorbital canal, and supraorbital bar, also develop by twisting slightly in the same direction as that in which the eye migrates [81]. Interestingly, the present results showed that the transitions from first to second phase in terms of total

filament number were mostly in agreement with the occurrence of metamorphic events, at least in the early larval stage, because an increase in slope was observed at almost the same body size (7.71 mm in body length; 0.005 g in body mass) (Figure 3A and Table 1). Thus, the present results suggest that differences in the skeletal structures of bilaterally symmetric and asymmetric fish can be used to explain differences in the ontogenetic process in gill morphology, at least during the early developmental stages of such fish.

However, such conclusions should be drawn with caution because the quantification of gill measurements here was based on restricted populations and/or measurement techniques. Previous studies have shown that the relationships of body mass to GA and/or the filament number in common carp were incongruent among relationships reported in previously published papers [74,88,89]. For fish weighing 184, 531, or 878 g in the study of Oikawa and Itazawa [74], the GA was slightly larger than that reported by Saunders [88] or Hughes and Morgan [89], whereas the filament number was considerably smaller. In future studies, it will be necessary to compare different populations and/or different measurement techniques to confirm whether differences exist between bilaterally symmetric and asymmetric fish.

*4.2. Body Morphometry*

Differences between bilaterally symmetric and asymmetric fish were also observed in the total cutaneous area and body surface area (Figure 4A,B and Table 2). In red porgy, the total cutaneous area is known to rapidly increase with positive allometry (slope of 3.986) during the early larval stage (at 0.00020–0.00025 g) and then slightly increase with diphasic negative allometry (slopes of 0.562 and 0.652, respectively) during larval (0.00028–0.00450 g) and later (0.0045–1230.0000 g) stages [75]. Although the changes in slope following an increase in body mass do not correspond to the total cutaneous area, these patterns were also found in relation to body surface area [75]. In contrast, the present results showed that the relationship between total cutaneous area had triphasic negative allometry, but the slopes quickly decreased from 0.815 to 0.625 when the phasic transition occurred from first to second phase during the larval to juvenile stage (at 0.00555–0.11665 g; Figure 4A and Table 2). Similar patterns were also observed for body surface area (Figure 4B and Table 2). These observations are consistent with the onset of metamorphosis, as mentioned above (at 7.71 mm in body length and 0.005 g in body mass) [81].

These differences in the slope patterns for total cutaneous area and body surface area that exist between bilaterally symmetric and asymmetric fish could be attributable to differences in their geometric type. Although it is difficult to directly measure body surface area, particularly in the early development stages, because of the extremely small size of the fish being measured, Hirst et al. [45], Glazier et al. [51], and Tan et al. [57] achieved this indirectly using derivations from mass–length scaling relationships in diverse aquatic invertebrates, which is a method based on simple Euclidean geometry. According to these studies, the scaling slopes of body surface area (as well as metabolic rate) are positively correlated with the degree of body shape, as predicted by the SA theory, but in contradiction to the negative correlations predicted by the RTN theory [45,51,57]. In particular, the spiny lobster (*Sagmariasus verreauxi*) exhibits a remarkable change in body shape from flattened two-dimensional growth in phyllosoma larvae to isometric three-dimensional growth in juveniles [51]. According to estimates of body surface area obtained using the aforementioned method, the scaling exponents of body surface area decreased from 0.91 in the phyllosoma larvae to 0.67 in the juveniles [51]. Likewise, the scaling exponents of routine metabolic rate also decreased from $1.002 \pm 0.081$ (95% CI) in the phyllosoma larva to $0.829 \pm 0.157$ in the juveniles. Therefore, as expected, the SA theory also predicts a positive correlation between the scaling exponents for body surface area and metabolic rate. This is a good example of how changes in body shape can have a positive effect on the scaling exponent of body surface as well as the metabolic rate of aquatic invertebrates.

In the present study, the slopes of both the total cutaneous area and body surface area in the larval to early juvenile stages (i.e., during metamorphosis) were close to 2/3, whereas the slopes in the later stage were >2/3 ($p < 0.01$; Figure 4A,B and Table 2). Interestingly, these changes were precisely corrected with the ontogenetic shifts in body shape because the larvae to early juveniles showed isometric three-dimensional growth, whereas the juveniles postmetamorphosis showed flattened two-dimensional growth. Contrastingly, in red porgy, growth of equal proportions in all three dimensions has been recorded; thus, either the slopes of the total cutaneous area and body surface area in the late larval stage were <2/3 ($p < 0.001$ and $p < 0.05$, respectively) or the slopes in the later stage were close to 2/3 [75]. Similar tendencies have been reported for other bilaterally symmetric fish, including herring, black sea bream, and common carp [68,74,90]. Therefore, it is likely that rapid ontogenetic changes in body shape, particularly in asymmetric fish, strongly affect the scaling of body surface area relative to that in bilaterally symmetric fish. Furthermore, even in smaller asymmetric fish (i.e., bilaterally symmetric larvae), the scaling of metabolic rate is expected to be greater than that in bilaterally symmetric fish because the correlations between the scaling exponents for body surface area and metabolic rate are also positive (if SA theory also applies in this case). When comparing the ratio of mass-specific areas of the gill surface to the body surface at the larval stage (i.e., at approximately 0.002 g; Table 4), the present results for the Japanese flounder indicate that the ratio of the mass-specific body surface is 6.67 times higher than that of the gill surface, although the initial body mass is almost identical. This finding supports the previous metabolic scaling theory based on the influence of body surface, because the ontogenetic changes in body shape are positively correlated with the scaling exponents of body surface area, even when asymmetric and symmetric fish are compared.

**Table 4.** Comparing the ratios of mass-specific areas of the gill surface to those of the body surface in bilaterally symmetric and asymmetric fish during the early development stages (0.002 g in body mass).

| Species | Mass-Specific Areas of Gill Surface (mm$^2$) | Mass-Specific Areas of Body Surface (mm$^2$) | Ratio | Ref. |
|---|---|---|---|---|
| Red porgy | 362.74 | 9156.83 | 25.24 | [75] |
| Japanese flounder | 50.56 | 8507.87 | 168.26 | Present study |

Despite the experimental limitation of the current study, i.e., the small number of individuals tested, the body shape changes in asymmetric fish, compared with those in bilaterally symmetric fish, seem likely to play an important role as scaling exponents of body surface area that help meet the metabolic requirements of the body during early developmental stages.

*4.3. Relationship between Respiratory Area and Metabolism*

For resting metabolism in fish, the overall scaling exponent is expected to be proportional to a 0.85 power of body mass [38,48,50,91]. This finding has been widely observed across many bilaterally symmetric fishes such as common carp, rainbow trout, red porgy, tiger puffer (*Takifugu rubripes*), beluga (*Huso huso*), and sterlet (*Acipenser ruthenus*) [4,74,92–94]. Although few studies of metabolism in asymmetric fish have been conducted, a value of 0.83 has been reported for the larval and juvenile Japanese flounder with body masses of 0.0002–0.9000 g [59]. This suggests that the scaling exponent of 0.85 during early life stages may not be specific to various fish species.

Previous studies on metabolism in bilaterally symmetric fish have indicated that the GA may not directly regulate resting metabolism, at least during the larval stage. In carp, Winberg et al. [12] found that the resting metabolic scaling exponent was close to 0.84 in fish that weighed from 0.00032 to 3500 g. Similarly, Itazawa and Oikawa (in preparation) found that the scaling exponent was close to 0.83 in fish that weighed from 0.0019 to 620 g [74]. Thus, the scaling component, for carp at least, is approximately 0.83–0.84. In contrast, the

GA was proportional to 7.066 and 1.222 powers of body mass during the pre- and postlarval stages (0.0016–0.3300 g), respectively, and it was proportional to a 0.794 power of body mass during the juvenile and adult stages (0.33–2250.00 g) [74]. Except for the third phase of the GA, these slopes differed significantly from the slope of the relationship between resting metabolism and body mass ($p < 0.01$). These findings are consistent with results reported for salmon and trout, i.e., that the slope for the gill area-body mass relationship is larger than that for the resting metabolism relationship [95]. Similar to the results for most fishes, the present results show that the GA was proportional to a 2.197 power of body mass during the early larval stage (at 0.00200–0.01125 g) and to 1.354 and 1.114 powers of body mass during the larval and juvenile stages (0.01000–1.30738 g), respectively (Figure 2 and Table 1). Although further research is required, these findings indicate that the mass-specific resting metabolic rate may be independent of the GA during the larval and juvenile stages, even in asymmetric fish, because the slopes for the GA differed significantly from those for the resting metabolism–body mass relationship ($p < 0.01$).

Metabolic scaling has been linked to the scaling of body surface area in cutaneous breathing aquatic invertebrates but not in aquatic arthropods with hard exoskeletons, which largely prevent cutaneous respiration [45,51,57]. Thus, in aquatic amphipod crustaceans, the scaling of metabolic rate is more closely related to the scaling of gill surface area [76,96]. Although the resource supply and demand of fishes differs substantially from that of various aquatic invertebrates due to their contrasting growth patterns, locomotion costs, predation pressures, and lifestyle [57], the resting metabolic scaling of Japanese flounder, at least, might be more closely related to total cutaneous area than to gill surface area.

Except for the second phase ($p < 0.01$), the slopes of the total cutaneous area–body mass relationship from the present study were almost identical to those for the resting metabolism–body mass relationship of intact Japanese flounder (0.831 was previously reported by Yagi et al. [59]). Thus, the ontogenetic shifts in resting metabolic scaling could be linked to developmental changes in body shape and surface area scaling, which would be expected given the SA theory but would contradict the negative correlations predicted by RTN theory. Because juvenile Japanese flounder (i.e., postmetamorphosis individuals) grow to be flattened in two dimensions, they provide a good example of how changes in body shape correspond with SA theory. When an organism begins to grow in a flattened two-dimensional from, SA theory predicts that the scaling of their metabolic rate should be increased (isometrically to body mass, i.e., "b" is equal or close to 1) [45,53], whereas the RTN theory predicts the opposite (decreasing) result (negative allometry with body mass, i.e., "b" is < 2/3) [37,39,40,45]. As predicted by the SA theory, the slope for the total cutaneous area–body mass relationship for the juvenile stage (i.e., postmetamorphosis) in the third phase (at 0.11035–1.30738 g) was similar to that for resting metabolism–body mass relationship (Figure 4A and Table 2). Therefore, this result theoretically and practically supports the hypothesis that a change in body shape influences resting metabolic scaling in asymmetric fish.

The difference between the slope for the total cutaneous area–body mass relationship and that for the metabolism–body mass relationship in the second phase (at 0.00555–0.11035 g) was consistent with the onset of metamorphosis (at 7.40 mm in body length and approximately 0.005 g in body mass according to Okada et al. [81]; Figure 4A and Table 2). Likewise, the present results showed that the slopes for total length gradually increased, except during the metamorphic portion of the second phase (at 0.00555–0.08122 g; Figure 4C and Table 2). It is reasonable to suggest that the rapid body shape changes are accompanied by a shift in metabolism because the ontogenetic phase shift in metabolism during the transformation from larva to juvenile occurs when an individual reaches a mass of approximately 0.01 g [59]. Specifically, this metamorphic transformation involves a process of changing body shape from an isomorphic three-dimensional form to a flattened two-dimensional form. Thus, the larval and/or early juvenile Japanese flounder (i.e., individuals undergoing metamorphosis) maintains isomorphic three-dimensional growth, because they have not yet completely transformed their body shape into the flattened two-dimensional form. In this case, the SA

theory predicts that the scaling of metabolic rate should be decreased (showing negative allometry with body mass, i.e., "b" is close to 2/3) [51], whereas the RTN theory conversely predicts that scaling should be increased (showing negative allometry with body mass, i.e., "b" is >2/3) [37,39,40,54] depending on network properties [39,40,42,55,56]. Given that ontogenetic shifts in resting metabolic scaling and those of body surface area are positively correlated [45,51,57], the decrease in the slope of the total cutaneous area–body mass relationship should correspond exactly with that of the metabolism–body mass relationship predicted by the SA theory, because its slope was close to 2/3 in the second phase (at 0.00555–0.11035 g; Figure 4A and Table 2). Furthermore, it is possible that metamorphic duration, i.e., the time over which the bilaterally symmetric to asymmetric change in form occurs, leads to changes other than those affecting swimming behavior. Fukuhara [97] found that during the transformation from the larval to juvenile stage in Japanese flounder, fish showed a relatively sharp increase in maximum speed, suggesting that they swim much faster during metamorphosis than during either the larval or juvenile stages. Previous studies have suggested that gill respiration has a relatively stronger association with active metabolism than with resting metabolism [74,85,98–107]; hence, the discrepancy between the slopes of the total cutaneous area–body mass relationship and the resting metabolism–body mass relationship for the metamorphic duration in the second phase (at 0.00555–0.11035 g; Figure 4A and Table 2) might be related to the increase in mobility that accompanies the associated morphological and behavioral changes. Therefore, according to the present study, there is a possibility that body surface plays an important role in gas exchange during the early developmental stage, at least in relation to the resting metabolic state. In contrast, it has previously been reported that the slopes of the total cutaneous area–body mass relationship in bilaterally symmetric fish during the early developmental stage significantly differ from those of the resting metabolism–body mass relationship during the larval to adult stage [74,75]. Although the current findings are unlikely to be representative of all asymmetric fish, the results related to body surface during the early developmental stage (excluding metamorphosis) provide evidence of a relationship with resting metabolism, at least for Japanese flounder.

　　In conclusion, the morphometric study presented here provides one example of how resting metabolic scaling is closely related to body shape and body surface in asymmetric fish. In particular, the resting metabolic scaling of Japanese flounder is likely associated more with total body (cutaneous) surface area than with gill surface area, because the slope for the total cutaneous area–body mass relationship was almost the same as that for the resting-body mass relationship during the early stages of development. Furthermore, the scaling of surface area with body mass in this species was significantly positively correlated with body shape during ontogeny. Importantly, this was predicted by the SA theory, but the result contradicted the negative correlations predicted by the RTN theory; specifically, changes to body shape and the scaling of surface area were almost identical to those predicted using the SA theory. Therefore, this study provides a good example of how changes in body shape influence metabolic scaling in asymmetric fish. Further studies, which must include more individuals and species, should examine the correlation between body surface and resting metabolism to ensure that the current results are applicable to other types of asymmetric fish.

**Funding:** This research received no external funding.

**Institutional Review Board Statement:** This study was conducted according to the guidelines of Animal Experiments in Faculty of Agriculture in Kyushu University (Fukuoka, Japan).

**Informed Consent Statement:** Not applicable.

**Data Availability Statement:** Data generated in this study are available on request from the corresponding author.

**Acknowledgments:** I would like to thank Dr. Shin Oikawa, former professor in the Fishery Research Laboratory, Kyushu University (Fukuoka, Japan), for the support and suggestions he provided when

the experiments were being performed. I would also like to acknowledge Tsujigawa Masaaki and Kodo Chen for their help in maintaining the experimental fishes. I also wish to thank Dr. Keitaro Kato and Dr. Youhei Washio in the Aquaculture Research Institute, Kindai University (Wakayama, Japan), for their support and consideration. In addition, I wish to thank the editors providing English language editing. Finally, I sincerely thank the Editor and the three anonymous Reviewers for their careful reading of the manuscript and their constructive comments and insightful suggestions.

**Conflicts of Interest:** The author declares no conflict of interest.

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
