# Peer review of "Ontogeny of the Respiratory Area in Relation to Body Mass with Reference to Resting Metabolism in the Japanese Flounder, Paralichthys olivaceus (Temminck & Schlegel, 1846)"

_fishes, doi:10.3390/fishes7010039_

Round 1

Reviewer 1 Report

This paper reports scaling of respiratory areas in the Japanese flounder and compares with scaling of metabolic rate. The experiment designs are good and the results could contribute in the biological scaling theory. I think it has values to be published. However, the paper writing need major revisions, especially for the introduction and discussion sections. The author needs to make more efforts to present the scaling mechanisms of area as well as metabolic rate.

Major comments

  1. This paper did not mention any scientific hypothesis around metabolic scaling and respiratory areas. The author wants to keep in mind that something ‘has yet to be demonstrated.’ would not be a good reason to start a study. I suggest that the introduction section include more details of the mechanisms on how areas correlation with metabolic rate and scaling, and then propose a suitable hypothesis. Since no hypothesis was proposed, most context of the discussion section make no sense. The author simply repeats the results and compare with previous reports. More information should be talked on the scaling mechanisms of body area and gill area and their links with the scaling of metabolic rate. Again, this needs a good hypothesis.
  2. This word did not measure metabolic rate. Metabolic rate of fish varies dependent of a lot of environmental factors. The author should make sure the previous reported data of metabolic rate they cited obtain in the same condition as in their work.

Minor comments

L82. How did you weigh the fish as the fish size was very small. Describe the equipment.

L94. Did you sample several filaments or measure all filaments?

L97. Again. Did you measure spaces in all filaments or just those sampled.

L139. Add a section of data analysis. Introduce any statistical methods, sofwares you used and how the results were presented.

L143-150. Did you observe the gills morphology every day? Or just at 11, 17, 30, 35 days post breeding. Nothing was mentioned in the section of methods.

Figure 1. Did you present mean values of just data of one individual?

L162. P<0.05. Present exact p value rather than just present p<0.05. The same problems also appeared several time in other places.

Figure 2 and Table 1. Make sure you adopt decimal places in right way. Some data was presented with 5 decimals. I don’t know how you can obtain such accurate measurements, which is up to the methods and equipment.

Table 1. M would be better abbreviation for body mass while W generally represents body weight.

Figure 2, 3, 4. You can differ the three stages by using different spot types.

Figure 5. Only two growth stages.

L270-273. More logically on how your results ‘could potentially be used to explain differences between the metamorphic characteristics of bilaterally symmetric and asymmetric fish’. You can not just say they happen at the same time.

L316-318. The asymmetric fish has different geometric type and then body surface area scaling compared those of bilaterally symmetric fish. Read the following papers and reconstruct this part of discussion.

Glazier DS, Hirst AG, Atkinson D. 2015 Shape shifting predicts ontogenetic changes in metabolic scaling in diverse aquatic invertebrates. Proc. R. Soc. B 282: 20142302.

Hirst AG, Glazier DS. and Atkinson D 2014 Body shape shifting during growth permits tests that distinguish between competing geometric theories of metabolic scaling. Ecology Letters, (2014) 17: 1274–1281

Tan H, Hirst AG, Glazier DS, Atkinson D. 2019 Ecological pressures and the contrasting scaling of metabolism and body shape in coexisting taxa: cephalopods versus teleost fish. Phil. Trans. R. Soc. B 374: 20180543.

L352-356. When you compare the scaling of GA to that of metabolic rate, you must make sure the fish have similar body size ranges. The scaling slope of metabolic rate also has a larger value during the post-larval stages. You cannot compare with the slope of 0.83 in fish weighing 0.0019–620.0000 g.

L390. A clear conclusion is needed at the end of discussion.

Author Response

Dear Reviewer 1:

I appreciate the precious time you have given to review my manuscript and the valuable comments you provided. Because of these insightful comments, I was able to greatly improve the manuscript. I have carefully considered each of your comment and made every effort to address each one. I apologize for any inconvenience caused by the unexpectedly time-consuming revision. I have strengthened the historical context of the study according to the comments and cited many additional papers to providing suitable hypotheses. The discussion section has been extensively revised to match the revised introduction section. Nevertheless, the main finding of the manuscript, i.e., body surface likely plays an important role in the regulation of resting metabolism before and after metamorphosis in Japanese flounder, remains unchanged. I have additionally described the relationship between body shape and metabolic scaling because body surface area is strongly affected by changes in body shape (e.g., metamorphosis). This detail has been added to the revised manuscript based on your valuable literature recommendations. I have provided my point-by-point responses to your comments below. All modifications in the manuscript are highlighted in red. Please see the attachment.

Thank you again for your kind consideration of the manuscript.

Sincerely,

Dong-In Kim

Aquaculture Research Institute, Kindai University, Shirahama 3153, Nishimuro, Wakayama 649-2211, Japan

Phone No: +81-80-4691-7627

Email Address: donginkim0508@gmail.com, dongin_kim@kindai.ac.jp

Reviewer 2 Report

General comments:

This study is remarkable for examining the scaling of respiratory surface area (for both gills and total body surface) across more than 3 orders of magnitude of body mass in a fish (including tiny larvae to large adults).  This analysis is very valuable for testing whether the scaling of metabolic rate is related to respiratory surface area.  However, I have only one major concern.  The author does not place his study in proper historical context because he fails to cite several papers that have examined metabolic scaling in relation to respiratory (gill or total body) surface area in animals generally. Nor does he acknowledge that numerous studies have examined ontogenetic shifts in metabolic scaling.  Also see specific comments.

Specific comments:

L 26-29:  Although metabolic rate may affect life histories, the reverse may also occur (see Glazier 2015). 

L 36-37:  The author may wish to read Glazier (2014) who reviews the major mechanisms proposed to explain metabolic scaling. 

L 38-45:  Please clarify. Oikawa and colleagues analyzed the scaling of respiratory surface area (SA) over much of the ontogeny of fishes, from larvae to adults.  Glazier & Paul (2017) also examined the scaling of gill SA encompassing all postembryonic life stages of a freshwater amphipod.  Other studies that have examined the scaling of respiratory (gill) SA over large ranges of sizes and ages: e.g., Li et al. (2018), Luo et al. (2020) and Scheuffle et al. (2021), among others.  None of these studies are cited.  Scheuffle et al. (2021) provide a nice review.

L 45-46: Yes, this is true because it is difficult to estimate total body surface area.  However, Hirst et al. (2014), Glazier et al. (2015) and Tan et al. (2019) have done this indirectly by using derivations from mass-length scaling relationships, based on simple Euclidean geometry.  Older metabolic scaling studies testing the “surface law” have also estimated body surface area: e.g., Ellenby (1937), Edwards (1958), etc. Also see Ultsch (1976).

L 48-50: A good review of ontogenetic changes in metabolic scaling in fishes is Post & Lee (1996).   A good review of ontogenetic changes in metabolic scaling in animals generally is Glazier (2005). 

L 342: Change “the the” to “the”.

L 357-390: The author makes a plausible case that the body-mass scaling of metabolism in the Japanese flounder is more related to total body (cutaneous) SA than to gill SA.  This makes sense because cutaneous respiration plays an important role in fish during early development when the gills are not fully developed.  To place these results in context, the author may also wish to cite Hirst et al. (2014), Glazier et al. (2015) and Tan et al. (2019) who show that metabolic scaling parallels the scaling of body surface area in cutaneous breathing aquatic invertebrates, but not in aquatic arthropods with hard exoskeletons that largely prevent cutaneous respiration.  Thus, in aquatic amphipod crustaceans, the scaling of metabolic rate is more related to the scaling of gill surface area (see Glazier & Paul 2017; Glazier et al. 2020).   

Literature cited:

Edwards, R. W. (1958). The relation of oxygen consumption to body size and to temperature in the larvae of Chironomus riparius Meigen. Journal of Experimental Biology35(2), 383-395.

Ellenby, C. (1937). Relation between body size and metabolism. Nature140(3550), 853-853.

Glazier, D. S. (2014). Metabolic scaling in complex living systems. Systems2(4), 451-540.

Glazier, D. S. (2015). Is metabolic rate a universal ‘pacemaker’ for biological processes? Biological Reviews90(2), 377-407.

Glazier, D. S., Hirst, A. G., & Atkinson, D. (2015). Shape shifting predicts ontogenetic changes in metabolic scaling in diverse aquatic invertebrates. Proceedings of the Royal Society B: Biological Sciences282(1802), 20142302.

Glazier, D. S., & Paul, D. A. (2017). Ecology of ontogenetic body-mass scaling of gill surface area in a freshwater crustacean. Journal of Experimental Biology220(11), 2120-2127.

Glazier, D. S., Borrelli, J. J., & Hoffman, C. L. (2020). Effects of fish predators on the mass-related energetics of a keystone freshwater crustacean. Biology9(3), 40.

Hirst, A. G., Glazier, D. S., & Atkinson, D. (2014). Body shape shifting during growth permits tests that distinguish between competing geometric theories of metabolic scaling. Ecology Letters17(10), 1274-1281.

Li, G., Lv, X., Zhou, J., Shen, C., Xia, D., Xie, H., & Luo, Y. (2018). Are the surface areas of the gills and body involved with changing metabolic scaling with temperature? Journal of Experimental Biology221(8), jeb174474.

Luo, Y., Li, Q., Zhu, X., Zhou, J., Shen, C., Xia, D., Djiba, P.K., Xie, H., Lv, X., Jia, J. and Li, G., 2020. Ventilation frequency reveals the roles of exchange surface areas in metabolic scaling. Physiological and Biochemical Zoology93(1), pp.13-22.

Post, J. R., & Lee, J. A. (1996). Metabolic ontogeny of teleost fishes. Canadian Journal of Fisheries and Aquatic Sciences53(4), 910-923.

Scheuffele, H., Jutfelt, F., & Clark, T. D. (2021). Investigating the gill-oxygen limitation hypothesis in fishes: intraspecific scaling relationships of metabolic rate and gill surface area. Conservation Physiology9(1), coab040.

Tan, H., Hirst, A. G., Glazier, D. S., & Atkinson, D. (2019). Ecological pressures and the contrasting scaling of metabolism and body shape in coexisting taxa: cephalopods versus teleost fish. Philosophical Transactions of the Royal Society B374(1778), 20180543.

Ultsch, G. R. (1976). Respiratory surface area as a factor controlling the standard rate of O2 consumption of aquatic salamanders. Respiration Physiology26(3), 357-369.

Author Response

Dear Reviewer 2:

I appreciate the precious time you have given to review my manuscript and the valuable comments you provided. Because of these insightful comments, I was able to greatly improve the manuscript. I have carefully considered each of your comment and made every effort to address each one. I apologize for any inconvenience caused by the unexpectedly time-consuming revision. I have strengthened the historical context of the study according to the comments and cited many additional papers to providing suitable hypotheses. The discussion section has been extensively revised to match the revised introduction section. Nevertheless, the main finding of the manuscript, i.e., body surface likely plays an important role in the regulation of resting metabolism before and after metamorphosis in Japanese flounder, remains unchanged. I have additionally described the relationship between body shape and metabolic scaling because body surface area is strongly affected by changes in body shape (e.g., metamorphosis). This detail has been added to the revised manuscript based on your valuable literature recommendations. I have provided my point-by-point responses to your comments below. All modifications in the manuscript are highlighted in red. Please see the attachment.

Thank you again for your kind consideration of the manuscript.

Sincerely,

Dong-In Kim

Aquaculture Research Institute, Kindai University, Shirahama 3153, Nishimuro, Wakayama 649-2211, Japan

Phone No: +81-80-4691-7627

Email Address: donginkim0508@gmail.com, dongin_kim@kindai.ac.jp

Reviewer 3 Report

I am glad to have participated in the editorial process of such an interesting manuscript. The subject matter is certainly niche, it concerns mainly the anatomy and physiology of aquatic animals, but the study is of considerable importance and I found it conducted impeccably with some key results obtained. In addition, the manuscript has a high linguistic quality, and is very accurate and detailed in all its parts, I compliment this work done alone, as I recognize the difficulties.

Moreover, I appreciated how the Author mentioned the limitations of this study (for example the small number of specimens studied) and how he proposed further effects of this study through subsequent research, with an important knowledge base resulting from this. Every research should, in my opinion, go through these critical and growth stages, well done.

For all these reasons, my opinion on this research product is good and I have only some minor suggestions to give to help the Author to improve it.

Title

Please add (Temminck & Schlegel, 1846) following the name of the species, for a more correct way of exposing it.

Abstract

The manuscript is very accurate and complex in its parts, almost verbose in some, but it can fit. The Abstract section, however, is in the present form a little too rich in details that disperse the attention of the reader and weigh it down a bit, for example the part between brackets in the first line, and all similar. Please try to be more discursive and light in this section, reserving accurate details for the contents of the manuscript.

Keywords

The common name of the species is already reported in the Title. To give more soundness in the web searches of other researches, is better to avoid this kind of repetition. Try tu substitute it with some related one, such as: fish morphology; teleost; asymmetric fish.

Material and Methods

Statistical analysis performed in this study, the results of which are reported in the Results section and discussed subsequently, were not reported and mentioned in this section. Please add a separate paragraph 2.7 to report the description of the methods and tests used.

Best regards

The Reviewer

Author Response

Dear Reviewer 3:

I appreciate the precious time you have given to review my manuscript and the valuable comments that you provided. Because of these insightful comments, I was able to greatly improve the manuscript. I carefully considered each of your comments and made every effort to address each one. I apologize for any inconvenience caused by the unexpectedly time-consuming revision. As you suggested, I added a statistics subsection to the methods section. I have also strengthened the historical context of the study according to the comments, and I have cited many additional papers when providing suitable hypotheses. The discussion section has been extensively revised to match the revised introduction section. Nevertheless, the main finding of the manuscript, i.e., that “body surface likely plays an important role in the regulation of resting metabolism before and after metamorphosis in Japanese flounder,” remains intact. The abstract section has been reorganized, without using parentheses, to make it more understandable for readers. Below, I provide my point-by-point responses to your comments. All modifications in the manuscript are highlighted in red. Please see the attachment.

Thank you again for your kind consideration of the manuscript.

Sincerely,

Dong-In Kim

Aquaculture Research Institute, Kindai University, Shirahama 3153, Nishimuro, Wakayama 649-2211, Japan

Phone No: +81-80-4691-7627

Email Address: donginkim0508@gmail.com, dongin_kim@kindai.ac.jp
